# Efficacy of Dust and Wettable Powder Formulation of Diatomaceous Earth (Detech^®^) in the Control of *Tyrophagus putrescentiae* (Schrank) (Acari: Acaridae)

**DOI:** 10.3390/insects13100857

**Published:** 2022-09-20

**Authors:** Nihal Kılıç

**Affiliations:** Plant Protection Department, Agriculture Faculty, Tekirdağ Namık Kemal University, Campus St. Değirmenaltı, Tekirdağ 59030, Turkey; nkilic@nku.edu.tr

**Keywords:** *Tyrophagus putrescentiae*, diatomaceous earth, mortality, WP formulation, dust formulation

## Abstract

**Simple Summary:**

Acaricidal efficacy of two Turkish diatomaceous earth (DE) formulations (Detech^®^ WP95 and Detech^®^ Dust) were applied on a concrete surface at five different concentrations at 25 ± 1 °C and 75 ± 5% RH. Mite mortalities were observed after 6- and 18-hour exposure periods dust and WP formulations, respectively. In conclusion, our laboratory tests indicated that both WP and dust formulations of local diatomaceous earth can cause 100% mortality in 24 h on average and can be a promising alternative to conventional chemical acaricides.

**Abstract:**

*Tyrophagus putrescentiae* (Schrank) (Acari: Acaridae) is a cosmopolite mite species commonly in found food and stored products. In this study, the acaricidal activity of two Turkish diatomaceous earth (DE) formulations (Detech^®^ WP95 and Detech^®^ Dust) were applied on a concrete surface at five different concentrations (1, 2.5, 5, 7.5, and 10 g/m^2^) and dead individuals were counted at 11 different time intervals (1, 3, 6, 9, 12, 15, 18, 21, 24, 27, and 30 h) at a temperature of 25 ± 1 °C and 75 ± 5% relative humidity (RH). Mite mortalities were observed after 6- and 18-hour exposure periods at all concentrations of dust and wettable powder (WP) formulations, respectively. Specifically, 100% mortality for the WP formulation was achieved at the highest concentration of 10 g/m^2^ after 15 h of exposure and after 27 h and 30 h for the lowest concentration. In the case of dust formulation, mortalities were observed after 3 h of exposure at all concentrations except at 1 g/m^2^, while a 100% mortality rate was achieved after 21 h of exposure to all concentrations and after 18 h of exposure for 7.5 g/m^2^ and 10 g/m^2^. This study indicates that both WP and dust formulations of local diatomaceous earth can cause 100% mortality in 24 h on average and can be a promising alternative to conventional chemical acaricides.

## 1. Introduction

*Tyrophagus putrescentiae* is a common and harmful mite species encountered in many countries in grain, flour, and various food products, especially in stored products such as cheese and certain nuts that are high in fat and protein [1,2,3]. It is known that in addition to damaging the products by feeding on them directly, the mites contaminate these products with dead mites, skin residues, and feces, reducing the quality of the products. These mites also play a role in causing allergies or certain respiratory diseases [1,4]. Conventional chemicals and fumigants are commonly used to control *T. putrescentiae* on food products and in warehouses. Since the use of methyl bromide (MB) was phased out as a fumigant, some other alternatives have been used such as hydrogen phosphide (PH3) [5], sulfuryl fluoride (SF) [6,7], and extreme temperature applications [8,9]. Another limiting factor is that mites develop a resistance to some particular chemicals such as pirimiphos-methyl and fenitrothion [10,11]. Moreover, fumigants have limited activity during the egg stage of insects and mites. One alternative control tactic for insect pests is the use of diatomaceous earth (DE) as inert dust. DE products are already registered as grain protectants in many countries.

Diatomaceous earth is particularly used in the control of stored-product pests due to the easy removal of DE from the product to which it is applied, leaving no residue, and due to the fact that stored-product insect pests are less resistant to it. DE consists of natural frustules formed from the fossilized siliceous shells of diatomite algae. DE is composed of environmentally friendly natural components and has low toxicity to mammals and the environment [12,13]. Due to its many advantages, DE is recommended as an alternative to insecticide [14,15]. Different DE products are proven to have insecticidal activity against stored-product insect pests [16,17,18,19,20,21,22,23,24,25,26,27,28,29,30,31,32,33,34,35,36]. The insecticidal activity of diatomaceous earth depends on some physical and chemical properties such as its composition, geological and geographical origin, SiO_2_ content, pH, and tapped density [37,38]. DE kills soft-bodied organisms such as mites in a shorter exposure time and at a higher rate than insects [39,40]. However, to our knowledge, no study has investigated the efficacy of DE products against *T. putrescentiae*.

In this study, the acaricidal activity on *T. putrescentiae* is compared for the first time in different concentrations of wettable powder (WP) and dust formulations of the local Turkish diatomaceous earth (Detech^®^-Entoteam R&D Co., Tekirdağ, Turkey).

## 2. Materials and Methods

### 2.1. Mite Species

The *Tyrophagus putrescentiae* used in this study originated from laboratory stock cultures and were reared on a mixture of soft wheat (*Triticum aestivum* L.), dog food, and wheat bran (1:1:1) at Acarology Laboratory of Tekirdag Namık Kemal University for more than 5 years. Biological tests were carried out at a temperature of 25 ± 1 °C and a relative humidity (RH) of 75 ± 5% for 24 h under dark laboratory conditions.

### 2.2. DE Formulations

In the biological tests, a local DE formulation, Detech^®^ (Entoteam R&D Co., Tekirdağ, Turkey) was used. Detech^®^ consists of a mixture of three types of freshwater DE collected from different DE reserves located in different regions of Turkey and has been commercially registered as an animal food additive. Detech^®^ is a light gray earth of freshwater origin obtained from diatomite deposits and consists of three different diatom frustules [32]. Its dust formulation consists of 100% amorphous diatomaceous earth particles, and the wettable powder formulation (WP) contains 95% amorphous diatomaceous earth particles and 5% surfactant and wetting agents. According to the total quantitative chemical analysis of Detech^®^ by atomic absorption spectroscopy (AAS), it contains approximately 80.6% SiO_2_, 4.75% CaO, 4.7% Al_2_O_3_, 1.5% Fe_2_O_3_, 0.85% MgO, 0.5% K_2_O, 0.4% Na_2_O, and less than 0.01% TiO_2_. Particle size analysis was conducted using the laser light diffraction technique and its median particle diameter (d(0.5)) is 14.061 μm.

### 2.3. DE Surface Treatments

In the biological tests, a concrete surface was used. In terms of its similarity to the grain storages, each cell base is covered with a concrete surface. For the preparation of the concrete surface, the mortar consisted of a mixture of 50 g cement and 10 mL water, which was was prepared and poured into 14 mm diameter plexiglass quintuple cells. Thereafter, the mortar in the quintuple cells was allowed to dry for 24 h at room temperature. Quintuple mite cells recommended by Solomon and Cunnington [41] and modified by Emekçi and Toros [42] were used for the acaricidal effect tests (Figure 1).

Dust and wettable powder formulations of local diatomaceous earth (Detech^®^ WP 95 and Detech^®^ Dust) were applied to the mite cells on the concrete surface in five different concentrations (1, 2.5, 5, 7.5, 10 g/m^2^). After the DE treatments, dead and alive individuals were counted at 11 different time intervals (1, 3, 6, 9, 12, 15, 18, 21, 24, 27, and 30 h). The experiments were carried out at a temperature of 25 ± 1 °C, 75 ± 5% RH, and under dark laboratory conditions.

Each replication of the concrete surface in each 14 mm diameter cell was treated with 1 mL of distilled water (control treatment) or an aqueous suspension (1 mL) of Detech^®^ WP 95 to provide five different concentrations (1, 2.5, 5, 7.5, and 10 g/m^2^). A Detech^®^ WP 95 suspension was applied to the concrete surface in the cells using a 1000 ppm micro-pipette. Following the Detech^®^ WP 95 suspension treatments, the cells were left to dry for 24 h under laboratory conditions. For the dust treatment, the specified concentrations of Detech^®^ Dust (1, 2.5, 5, 7.5, and 10 g/m^2^) were calculated according to the surface area of the test cells, and the dust particles were applied directly to the concrete surface in the cells and dispersed uniformly using a soft brush.

For each DE formulation, two adult mites and a small piece of wheat bran as food were placed in each DE-treated cell. In order to prevent the mites from escaping, the top of the cell was covered with a cover glass (lamella) and fastened with forceps from the top (Figure 1). Each trial was replicated 10 times and 10 controls (water-treated and untreated surfaces) were left for each treatment. The DE-treated and untreated cells were placed in a desiccator where 75% humidity was provided with the appropriate amount of KOH solution [43] and the desiccator was taken to a dark laboratory with a temperature of 25 ± 1 °C. In order to determine the residual toxicity of the DE formulations, the dead and alive mites were counted and the dead ones removed under a stereo zoom microscope at 11 different time intervals (1, 3, 6, 9, 12, 15, 18, 21, 24, 27, and 30 hours) after the surface application.

### 2.4. Statistical Analysis

The experimental factors comprised concentrations at six levels and exposure time at 11 levels. Mortality data were analyzed by using two-way ANOVA [44] after applying Arcsin transformation separately to normalize the heteroscedastic treatment variances [45,46]. Differences between the means of the mortality rate were compared using the Tukey test at the 5% significance level. The lethal time to kill 50% (LT_50_) and 99% (LT_99_) of the mites were calculated, along with the corresponding 95% confidence intervals (CI) using POLOPC [47].

## 3. Results

A two-way analysis of variance (ANOVA) indicated that all main effects (concentration and exposure time interval) and their associated interactions for each DE formulation (Detech^®^ WP 95 and Detech^®^ Dust) were significant (Table 1).

Mite mortalities for Detech^®^ WP95 started at the lowest dose of 1 g/m^2^ after 18 h of exposure (Table 2) and complete mortality was reached after 27 h of exposure. Mortality began increasing gradually at the concentrations of 2.5, 5, and 7.5 g/m^2^, after 15, 12, and 9 h of exposure, respectively. At the highest concentration of 10 g/m^2^, mite mortality started after 3 h of exposure and 100% mortality was achieved after 15 h of exposure.

For other concentrations, 100% mortality was achieved after 27 and 30 h of exposure. The difference between the mortality rates obtained as of the 21st hour was found to be statistically insignificant (*p* < 0.05).

In the case of the Detech^®^ Dust formulation treatments, the mortalities of *T. putrescentiae* individuals started at 1 g/m^2^ concentration after 6 h of exposure, and after 3 h of exposure at all other concentrations (Table 3). A 100% mortality rate was achieved at 1, 2.5, and 5 g/m^2^ concentrations after 21 h of exposure, and at 7.5 and 10 g/m^2^ concentrations after 18 h of exposure. The mortality rates of Detech^®^ Dust at all tested concentrations after 18 h of exposure were 95% and above, and the differences were found to be statistically similar.

Comparing the efficacy of the Detech^®^ Dust and WP95 formulations, the dust formulation resulted in higher mortalities of the mites at lower concentrations and at earlier exposure times (Figure 2). Additionally, in the case of the Detech^®^ WP95 formulation, the mortalities of the mites only began at the highest concentration (10 g/m^2^) after 3 h of exposure. Moreover, we observed that in *T. putrescentiae* individuals, due to their soft body, desiccation started immediately after the DE treatments. The DE-treated and untreated (control) individuals are shown in Figure 3.

The analysis of the lethal time of *T. putrescentiae* adults exposed to Detech^®^ WP and Detech^®^ Dust diatomaceous earth formulations at five different concentrations is provided in Table 4. The lethal time (LT_50_/LT_95_/LT_99_) estimates for the Detech^®^ WP95 formulation applied to *T. putrescentiae* adults at the lowest concentration (1 g/m^2^) and highest concentration (10 g/m^2^) were 22, 27, and 34 h and 7, 14, and 24 h, respectively. For the Detech^®^ Dust formulation at the lowest concentration (1 g/m^2^) and highest concentration (10 g/m^2^), LT_50_, LT_95_, and LT_99_ were 15, 27, and 44 h and 8, 13, and 21 h, respectively.

## 4. Discussion

Many researchers reported in various studies that the efficacy of DE applications varies depending on some physical and chemical properties of the diatomaceous earth composition, such as its geological and geographical origin, SiO_2_ content, dose, formulation of the DE, tapped density, the temperature of the environment, and also the surface of the application area [21,27,36,37,38,39,40].

Successful applications, with the local diatomaceous earth formulations of Detech^®^, have also been performed against stored-product insects. Erturk et al. [27] investigated the efficacy of the wettable powder formulation of Detech^®^ on concrete and wooden surfaces at different concentrations against *Sitophilus oryzae* adults. They reported that, for the concrete surface treatments, mortalities started after 48 h of exposure at all concentrations of Detech^®^ WP and only reached 100% at the concentration of 2 g/m^2^ after 288 h of exposure. In another study, Saglam et al. [33] reported that a 900 ppm concentration of dust formulation of Detech^®^, which we also used in our experiments, caused 97%, 92%, and 61% mortality in *Tribolium confusum*, *Sitophilus oryzae*, and *Rhyzoperta dominica* after 7 days of exposure, respectively. Moreover, Detech^®^ was found to have high and moderate repellent properties to *T. confusum* and *S. oryzae* adults, respectively. 

Diatomaceous earth is used successfully to kill soft-bodied organisms such as mites in a shorter time and at a higher mortality rate than insects [16,17,18,22,23,39,40]. The storage mites *Lepidoglyphus destructor* and *Acarus siro* had 100% mortality through the application of a 0.5 g/kg concentration of the DE formulation, Dryacide [16]. Additionally, populations of the storage mites *T. putrescentiae*, *A. siro*, and *L. destructor* exposed to the DE formulation named Protect-It at 0.5, 1, 3, and 5 g/kg concentrations applied to canola seeds decreased more than 95% [19]. Although the mite species and DE used in our experiments are different, our results indicate that *T. putrescentiae* is also susceptible to lower concentrations of the local DE formulation, Detech^®^.

Another important factor for the efficacy of DE is the DE formulation. Collins and Cook [39] found that dust formulations of Silicosec and Diasecticide were more effective as a treatment, with mean mortalities ranging from 93% to 100% for certain mite species (*A. siro* and *L. destructor*) at 0.5, 1, and 2 g/m^2^ of the dry dusts in petri dishes in conditions of 15 °C and 80% RH. According to results of this study, *A. siro* reached 100% mortality in the first 24 h, while the corresponding rate was between 93% and 97% for the same three concentrations in *L. destructor*. Collins and Cook [40], in another study, tested the same DE against *L. destructor* at 1 and 3 g/m^2^ of the dry dust, which were assessed one day, 4 weeks, and 12 weeks after treatment on a wooden surface at conditions of 15 °C and 80% RH. Mortalities were evaluated after exposure times of 24 h and 7 days for the mites and the dry dust formulation was found to be more effective than the slurry formulation. Additionally, Silicosec represented 93–100% mean mortalities in a 24 h exposure time and was higher than the Diasecticide dry dust diatomaceous earth (68–93%). In our study, the dust formulation of Detech^®^ resulted in 100% mortality of *T. putrescentiae* adults on a concrete surface after 21 h of exposure, and our results were similar to those from Silicosec, but higher than those for the DE formulation Diasecticide.

The slurry application of DE is useful for preventing the very dusty atmospheres created by the dust application method. Exposure to dusty environments may cause various health problems. Moreover, dust formulations may cause abrasion on the tools and equipment used, or the use of special tools may be required [27,39]. Some researchers have already tested slurry forms of DE. Collins and Cook [39] tested Silicosec and Diasecticide applied as slurry at 2.5, 5, and 10 g/m^2^ against mite species *A. siro* and *L. destructor* in conditions of 15 °C and 80% RH. The researchers tested the same trademarked DE products at 20 g/m^2^ against only *L. destructor* under the same laboratory conditions for one day, 4 weeks, and 12 weeks after treatment [40]. Both study results concluded that the slurry application of DE was generally less effective than the dry dust at equivalent concentrations. In addition, *A. siro* was determined to be less tolerant than *L. destructor*. In our study, the results indicate that the Detech^®^ WP formulation is less effective than the dust formulation; 100% mortality of the mites was achieved at 15 h of exposure at the highest concentration (10 g/m^2^) of Detech^®^ Dust, while 15 h of exposure time with Detech^®^ WP95. Based on the above, the effectiveness of DE varies according to the concentration rate, composition of the formulation, and also the mite species.

Mite species is another important factor affecting the efficacy of DE applications. The same DE products result in different mortality rates according to the mite species. Palyvos et al. [20] reported that the predatory mite *Cheyletus malaccensis* is resistant to low concentrations of DE (0.5 g/kg of grain), and the mortality rate did not exceed 29% at the end of the exposure period of 24 h for the treated grain. The researchers suggested that low-dose DE application against these warehouse mites can be safely included in the IPM program, since it will preserve the presence of predator mites.

Diatomaceous earth has abrasive and water-absorbing properties, it causes death in mites by disrupting the structure of the waterproof components in the epicuticle layer of the cuticle [48,49]. *T. putrescentiae*, as astigmatid mites, do not have a tracheal system, so they breathe directly through their skin by means of diffusion; thus, the cuticle acts as a respiratory surface [1,50]. As astigmatic mites have a soft skin layer, as can be observed in our study, the tested DE products caused mortality by causing abrasion and drying of the mite skin. The dust formulation caused complete mortality after 21 h of exposure at all concentrations, including the lowest concentration of 1 g/m^2^. Our results indicate that *T. putrescentiae* is much more susceptible to the dust formulations of Detech^®^. This may be attributable to the fact that a certain number of mites (two adults per cell) are monitored in a very small area and the possibility of high exposure to diatom particles is enhanced.

The application surface is also very important for the effectiveness of DE products. Cook et al. [39] reported that Silicosec^®^ dry dust treatments on glass and plastic surfaces with a 0.5 g/m^2^ dose against *A. siro* and *L. destructor* mortality rates were determined as 100% and 95–100% in a 24 h exposure time for the mite species, respectively [39]. The same researchers tested Silicosec^®^ dry dust on a wooden surface at 15 °C and 80% RH against *L. destructor* for 24 h, 4 weeks, and 12 weeks, and found that mean mortalities of 93–100% were obtained over the 12 weeks [40]. Thus, wooden surface applications were found to be less effective than glass and plastic surface applications. Detech^®^ WP95 was found to be significantly more effective when applied to concrete surfaces than on wooden surfaces [27].

At high concentrations, adults of *T. putrescentiae* were killed more quickly than at low concentrations. The lethal time (LT_99_) estimates for Detech^®^ WP 95 at 1 g/m^2^ concentration were 1.6 times higher than those at a 10 g/m^2^ concentration; similarly, the lethal time estimates for Detech^®^ Dust at a 1 g/m^2^ concentration were two times higher than at a 10 g/m^2^ concentration.

## 5. Conclusions

In conclusion, our laboratory tests indicated that a 1 g/m^2^ concentration of Detech^®^ Dust can achieve satisfactory control of the warehouse mite *T. putrescentiae* after 21 h of exposure, on a concrete surface. The wettable powder formulation, Detech^®^ WP95, required 10 g/m^2^ for 21 h to obtain complete mortality. Obviously, the WP formulation of Detech^®^ needs a 10 times higher concentration than the dust formulation to obtain satisfactory control of the mites. Overall, this study indicates that both Detech^®^ formulations have high acaricidal efficacy against *T. putrescentiae* for a short period of 24 h on average and can be a promising alternative to conventional chemical acaricides. However, further research is required to evaluate its efficacy on different surfaces under laboratory and real-world conditions, and also its side effects on predatory mites.

## Figures and Tables

**Figure 1 insects-13-00857-f001:**
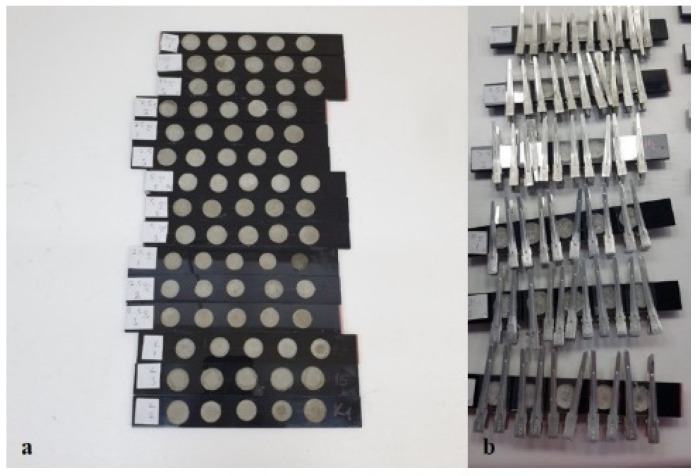
Plexiglass cells used in biological tests before (**a**) after (**b**) application.

**Figure 2 insects-13-00857-f002:**
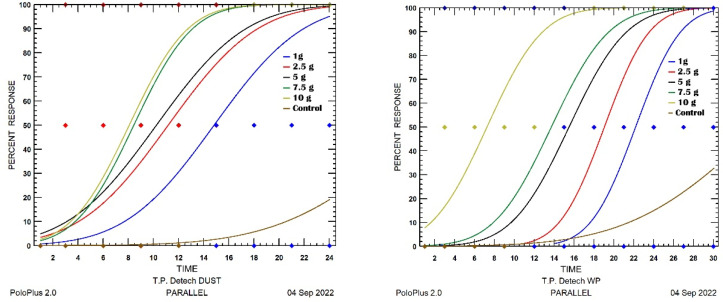
Dose–response (%) of *Tyrophagus putrescentiae* exposed to five different concentrations of Detech^®^ WP 95 and Detech^®^ Dust for certain observation times.

**Figure 3 insects-13-00857-f003:**
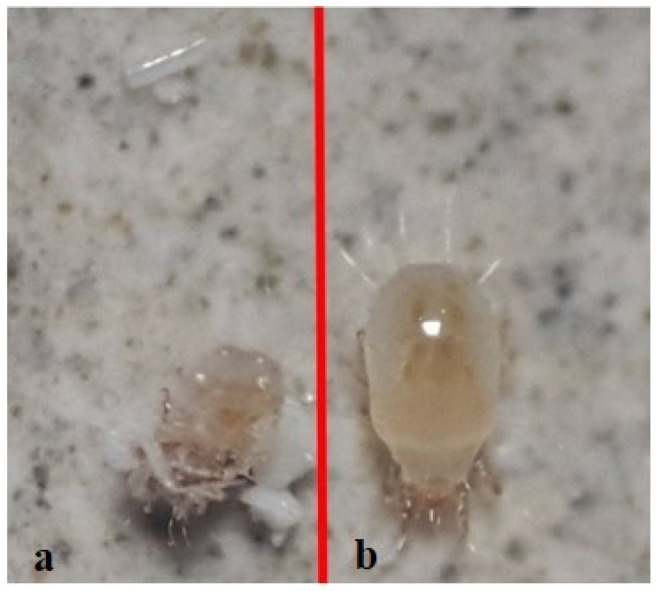
DE-treated (**a**) and untreated (**b**) *Tyrophagus putrescentiae* adults.

**Table 1 insects-13-00857-t001:** ANOVA parameters for main effects and associated interactions for mortality of *Tyrophagus putrescentiae* adults exposed to DE formulations of Detech^®^ WP 95 and Detech^®^ Dust.

	Tests between Main Effects					
	Detech^®^ WP			Detech^®^ Dust		
Source	df	F	*p*	df	F	Sig.
Corrected Model	65	52.110	<0.0001	65	59.217	<0.0001
Intercept	1	3368.310	<0.0001	1	6736.096	<0.0001
Time	10	167.404	<0.0001	10	240.055	<0.0001
Concentration	5	196.235	<0.0001	5	205.342	<0.0001
Time × Concentration	50	11.756	<0.0001	50	8.437	<0.0001
Error	594			594		
Total	660			660		
Corrected Total	659			659		

**Table 2 insects-13-00857-t002:** Average mortality rate (mean + SE) of *Tyrophagus putrescentiae* exposed to five different concentrations of Detech^®^ WP 95 for certain time periods.

	Mean Mortality Rate (%) ± SE											
Concentrations	1 h	3 h	6 h	9 h	12 h	15 h	18 h	21 h	24 h	27 h	30 h	F and *p* Values
**1 g/m^2^**	0 ± 0.0 * Ac	0 ± 0.0 Ac	0 ± 0.0 Bc	0 ± 0.0 Cc	0 ± 0.0 Ac	0 ± 0.0 Dc	15 ± 7.63 Cc	40 ± 10 Cb	55 ± 11,66 Bb	100 ± 0.0 Aa	100 ± 0.0 Aa	F_10,99_ = 60.198 *p* < 0.0001
**2.5 g/m^2^**	0 ± 0.0 Ad	0 ± 0.0 Ad	0 ± 0.0 Bd	0 ± 0.0 Cd	0 ± 0.0 Ad	15 ± 7.63 CDd	50 ± 10.54 Bc	70 ± 8.16 Bbc	90 ± 6.66 Aab	95 ± 5.0 Aa	100 ± 0.0 Aa	F_10,99_ = 67.369 *p* < 0.0001
**5 g/m^2^**	0 ± 0.0 Ac	0 ± 0.0 Ac	0 ± 0.0 Bc	0 ± 0.0 Cc	10 ± 6.66 Ac	45 ± 8.97 BCb	85 ± 7.63 Aa	95 ± 5.0 ABa	95 ± 5.0 Aa	100 ± 0.0 Aa	100 ± 0.0 Aa	F_10,99_ = 103.071 *p* < 0.0001
**7.5 g/m^2^**	0 ± 0.0 Ad	0 ± 0.0 Ad	0 ± 0.0 Bd	25 ± 8.33 Bcd	50 ± 12.90 Bbc	70 ± 13.33 ABab	85 ± 7.63 Aa	85 ± 7.63 ABa	95 ± 5.0 Aa	95 ± 5.0 Aa	100 ± 0.0 Aa	F_10,99_ = 32.684 *p* < 0.0001
**10 g/m^2^**	0 ± 0.0 Ac	15 ± 10.6 Ac	55 ± 15.72 Ab	70 ± 11.05 Aab	85 ± 7.63 Aab	100 ± 0.0 Aa	100 ± 0.0 Aa	100 ± 0.0 Aa	100 ± 0.0 Aa	100 ± 0.0 Aa	100 ± 0.0 Aa	F_10,99_ = 26.182 *p* < 0.0001
**Control**	0 ± 0.0 Ac	0 ± 0.0 Ac	0 ± 0.0 Bc	0 ± 0.0 Cc	0 ± 0.0 Ac	5 ± 5.0 Dc	5 ± 5.0 Cc	15 ± 7.63 Cbc	15 ± 7.63 Cbc	15 ± 7.63 Bab	35 ± 10.67 Ba	F_10,99_ = 9.789 *p* < 0.0001
**F and *p*** **values**	F_5,54_ = --- ---	F_5,54_ = 1.976 *p* < 0.097	F_5,54_ = 12.236 *p* < 0.0001	F_5,54_ = 25.174 *p* < 0.0001	F_5,54_ = 28.151 *p* < 0.0001	F_5,54_ = 27.995 *p* < 0.0001	F_5,54_ = 30.600 *p* < 0.0001	F_5,54_ = 21.941 *p* < 0.0001	F_5,54_ = 23.469 *p* < 0.0001	F_5,54_ = 63.923 *p* < 0.0001	F_5,54_ = 37.098 *p* < 0.0001	

* Means within rows followed by the same lowercase letter and within each column followed by the same uppercase letter are not significantly different (Tukey test at 5% level) (*p* < 0.05).

**Table 3 insects-13-00857-t003:** Average mortality rate (mean + SE) of *Tyrophagus putrescentiae* exposed to five different concentrations of Detech^®^ Dust for certain time periods.

Mean Mortality Rate (%) ± SE										
Concentrations	1 h	3 h	6 h	9 h	12 h	15 h	18 h	21 h	24 h	F and *p* Values
**1 g/m^2^**	0 ± 0.0 * Ac	0 ± 0.0 Ad	25 ± 8.33 Ac	30 ± 8.16 ABc	30 ± 8.16 CDc	60 ± 6.66 Cb	95 ± 5.00 Aa	100 ± 0.0 Aa	100 ± 0.0 Aa	F_8,81_ = 55.952 *p* < 0.0001
**2.5 g/m^2^**	0 ± 0.0 Ad	5 ± 5.0 Ade	20 ± 8.16 Ade	30 ± 8.16 ABcd	50 ± 10.54 BCbc	65 ± 7.63 BCb	95 ± 5.00 Aa	100 ± 0.0 Aa	100 ± 0.0 Aa	F_8,81_ = 41.616 *p* < 0.0001
**5 g/m^2^**	0 ± 0.0 Ac	5 ± 5.0 Ad	10 ± 6.66 Ad	45 ± 8.97 Ac	65 ± 7.63 ABbc	90 ± 6.66 ABab	95 ± 5.00 Aa	100 ± 0.0 Aa	100 ± 0.0 Aa	F_8,81_ = 59.130 *p* < 0.0001
**7.5 g/m^2^**	0 ± 0.0 Ad	15 ± 7.63 Ac	25 ± 8.33 Ac	55 ± 11.66 Ab	80 ± 8.16 ABab	95 ± 5.00 Aa	100 ± 0.0 Aa	100 ± 0.0 Aa	100 ± 0.0 Aa	F_8,81_ = 41.766 *p* < 0.0001
**10 g/m^2^**	0 ± 0.0 Ac	5 ± 5.0 Acd	30 ± 11.05 Ac	65 ± 10.67 Ab	90 ± 6.66 Aab	95 ± 5.00 Aa	100 ± 0.0 Aa	100 ± 0.0 Aa	100 ± 0.0 Aa	F_8,81_ = 48.498 *p* < 0.0001
**Control**	0 ± 0.0 Ac	0 ± 0.0 Aa	0 ± 0.0 Aa	0 ± 0.0 Ba	0 ± 0.0 Da	5 ± 5.00 Da	5 ± 5.00 Ba	15 ± 7.63 Ba	15 ± 7.63 Ba	F_8,81_ = 2.175 *p* = 0.038
**F and *p*** **values**	F_5,54_ = ---	F_5,54_ = 1.350 *p* = 0.258	F_5,54_ = 2.042 *p* = 0.087	F_5,54_ = 6.823 *p* < 0.0001	F_5,54_ = 19.310 *p* < 0.0001	F_5,54_ = 32.310 *p* < 0.0001	F_5,54_ = 85.000 *p* < 0.0001	F_5,54_ = 123.857 *p* < 0.0001	F_5,54_ = 123.857 *p* < 0.0001	

* Means within rows followed by the same lowercase letter and within each column followed by the same uppercase letter are not significantly different (Tukey test at 5% level) (*p* < 0.05).

**Table 4 insects-13-00857-t004:** Toxicity of Detech^®^ WP95 and Detech^®^ Dust formulations expressed as LT_50_, LT_95_, and LT_99_ values for *T. putrescentiae* adults after exposure to five concentrations.

Formulations	Concentration (g/m^2^)	N ^a^	Slope ± SE ^b^	t Ratio	H ^c^ (d.f.) χ2	Chi-square) ^d^	LT_50_ (h) (Confidence Intervals) ^e^	LT_95_(h) (Confidence Intervals) ^e^	LT_99_ (h) (Confidence Intervals) ^e^
Detech® WP95	1	110	0.31 ± 0.04	6.73	0.33	37.53	22.13 (21.10–23.20)	27.35 (25.84–29.92)	29.52 (27.58–32.93)
	2.5	110	0.25 ± 0.03	7.45	0.53	58.07	19.003 (17.84–20.14)	25.35 (23.71–27.90)	27.98 (25.89–31.38)
	5	110	0.36 ± 0.05	6.39	0.39	42.38	15.49 (14.67–16.31)	20.04 (18.88–21.84)	21.932 (20.45–24.31)
	7.5	110	0.17 ± 0.02	8.95	1.186	128.11	13.60 (12.04–15.10)	22.88 (20.69–26.17)	26.730 (23.91–31.11)
	10	110	0.22 ± 0.03	7.27	0.99	107.16	7.22 (5.94–8.42)	14.44 (12.90–16.71)	17.43 (15.11–21.28)
Detech® Dust	1	90	0.13 ± 0.01	7.49	0.90	79.30	15.13 (13.78–16.64)	27.18 (24.45–31.29)	32.17 (28.61–37.61)
	2.5	90	0.18 ± 0.02	8.20	0.83	73.35	11.15 (10.03–12.28)	19.98 (18.16–22.54)	23.63 (21.29–27.04)
	5	90	0.25± 0.03	7.70	0.65	57.76	10.24 (9.28–11.21)	16.82 (15.38–18.90)	19.55 (17.70–22.29)
	7.5	90	0.25 ± 0.03	7.42	0.56	49.56	8.41 (7.46–9.37)	14.92 (13.46–17.06)	17.61 (15.73–20.45)
	10	90	0.30 ± 0.04	7.09	0.71	62.55	8.05 (7.17–8.92)	13.50 (12.21–15.40)	15.75 (14.11–18.27)

^a^ Number of individuals treated, excluding controls, ^b^ t-ratio of the slope of regression test; t-ratio values greater than 1.96 indicate a significant regression, ^c^ heterogeneity factor equals the chi-square divided by the degrees of freedom; heterogeneity factor value less than 1.0 indicates that the log-dose–probit lines are within the 95% confidence limits and thus the model fits the data, ^d^ chi-square value, ^e^ numbers in brackets give the 95% confidence intervals.

## Data Availability

All data are contained within the article.

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
