# Peer review of "Efficacy of Dust and Wettable Powder Formulation of Diatomaceous Earth (Detech®) in the Control of Tyrophagus putrescentiae (Schrank) (Acari: Acaridae)"

_insects, 2022, doi:10.3390/insects13100857_

Round 1

Reviewer 1 Report

The manuscript entitled “Efficacy of dust and wettable powder formulation of diatomaceous earth (Detech®) to control of Tyrophagus putrescentiae (Schrank) (Acari: Acaridae)” it has valuable data that are of interest to readers of the Journal. The MS needs minor revision in order to be published in the journal.

In the attached file author can found my comments and suggestions that could help the improvement of the MS. 

Author Response

Dear Reviewer,

Many thanks for your time and valuable comments on the manuscript. I revised it as your suggestions. Please find comments and corrections on the MS file.

Yours sincerely

Nihal Kilic

Reviewer 2 Report

insects-1891196 Efficacy of dust and wettable powder formulation of diatomaceous earth

(Detech®) to control of Tyrophagus putrescentiae (Schrank) (Acari: Acaridae)

Authors: Nihal Kılıç.

In this manuscript, the authors conduct studies to evaluate efficacy of
diatomaceous earth formulations for controlling mites in stored products.  The results of this study are potentially valuable after the literature review defines clearly the new knowledge being sought in the study, and below comments are addressed.

Tables 1 & 3 require a heading and description. (as you did for table 2)

Data in tables 2 & 3 would be better presented in a line graph with an overall stat analysis like repeated measures of logistic regression.

Figure 2.  These comparison data also would be better in a line graph with stats. Current presentation is noisy and difficult to interpret.

Ln 30.  Replace “products, these” with “products. These ..”. new sentence.

Ln 36.  Replace “fumigants has a efficacy problems against to egg” with “fumigants have limited activity on egg stages…”

Ln 37. Replace “One alternative for insect pests are the use of diatomaceous” with “One alternative control tactic for insect pests is the use of diatomaceous”.

Ln 47 – 53. The authors states that “Different DE products are successfully used for the 46 control of stored-product insect pests and mites”. If so, what scientific knowledge is lacking, which will justify your study?  What is the problem you are intending to solve? This must be clearly listed, otherwise there is not potential for adding to the existing body of knowledge.

Ln 86.  What are the “cells”? (company source, city, state) need a reference.

Ln 120.  I suggest calculating “lethal time” using logistic regression or “repeated measures” to better show temporal differences between treatments.

Ln 176, I can’t fully review the discussion until previous comments are addressed.

Ln 177. Again, if you are making conclusions on “mortality rate”, then you should use more robust stat analysis.

Author Response

Dear Reviewer,

Many thanks for your time and valuable comments on the manuscript. I revised it as your suggestions and added Lethal time tables and dose-response graphs. Please find comments and corrections on MS file.

 Yours sincerely

Nihal Kilic

Round 2

Reviewer 2 Report

this manuscript will add new knowlege in this subject matter